# Skin Wound Healing: Normal Macrophage Function and Macrophage Dysfunction in Diabetic Wounds

**DOI:** 10.3390/molecules26164917

**Published:** 2021-08-13

**Authors:** Savannah M. Aitcheson, Francesca D. Frentiu, Sheree E. Hurn, Katie Edwards, Rachael Z. Murray

**Affiliations:** 1School of Biomedical Sciences and Centre for Immunology and Infection Control, Faculty of Health, Queensland University of Technology, Brisbane, QLD 4059, Australia; s.aitcheson@qut.edu.au (S.M.A.); francesca.frentiu@qut.edu.au (F.D.F.); 2School of Clinical Sciences, Faculty of Health, Queensland University of Technology, Brisbane, QLD 4059, Australia; sheree.hurn@qut.edu.au; 3School of Optometry and Vision Science, Faculty of Health, Queensland University of Technology, Brisbane, QLD 4059, Australia; katie.edwards@qut.edu.au

**Keywords:** macrophage, inflammation, diabetic foot ulcer, chronic wound, efferocytosis, phenotype, infection

## Abstract

Macrophages play a prominent role in wound healing. In the early stages, they promote inflammation and remove pathogens, wound debris, and cells that have apoptosed. Later in the repair process, they dampen inflammation and secrete factors that regulate the proliferation, differentiation, and migration of keratinocytes, fibroblasts, and endothelial cells, leading to neovascularisation and wound closure. The macrophages that coordinate this repair process are complex: they originate from different sources and have distinct phenotypes with diverse functions that act at various times in the repair process. Macrophages in individuals with diabetes are altered, displaying hyperresponsiveness to inflammatory stimulants and increased secretion of pro-inflammatory cytokines. They also have a reduced ability to phagocytose pathogens and efferocytose cells that have undergone apoptosis. This leads to a reduced capacity to remove pathogens and, as efferocytosis is a trigger for their phenotypic switch, it reduces the number of M2 reparative macrophages in the wound. This can lead to diabetic foot ulcers (DFUs) forming and contributes to their increased risk of not healing and becoming infected, and potentially, amputation. Understanding macrophage dysregulation in DFUs and how these cells might be altered, along with the associated inflammation, will ultimately allow for better therapies that might complement current treatment and increase DFU’s healing rates.

## 1. There Is a Clinical and Economic Need for Better Wound Therapies

Globally, 1 in 11 adults aged 20–79 has diabetes (463 million estimated cases); 25% of which will acquire a non-healing diabetic foot ulcer (DFU) in their lifetime [1,2,3]. Patients with DFUs tend to have decreased mobility, which can impact their ability to perform daily activities, leading to an increasing dependence on others [1]. This and other factors such as social isolation and depression negatively affect patients’ perceived health-related quality of life [4]. Most of these patients receive the standard care, which involves assessment of the wound, regular wound dressing changes, offloading, antibiotics if infected, and perhaps debridement to remove necrotic or infected tissue [1,5,6,7]. Some patients receive more advanced (and costly) therapies that are designed to help improve healing, including cell-based therapies such as Dermagraft^®^, a human fibroblast-derived dermal substrate designed to build up the granulation tissue, or Regranex, which is a platelet-derived growth factor therapy designed to attract cells to the wound [8,9,10]. However, many of these wounds still fail to heal and, if more severe tissue damage or an infection is not stemmed, then amputation is generally required [1,11].

Non-healing ulcers are the most common cause of amputation and it is estimated that, worldwide, a leg is amputated due to diabetes every 30 seconds [1,12,13]. Approximately 84% of lower leg amputations will be in patients that have acquired a DFU prior to amputation [2,5]. Of the DFUs that do not heal, approximately 25% will require amputation [1]. If further improvements in therapies are not made, the overall number of amputations will increase as the number of people with diabetes rises. The number of adults with diabetes has tripled over the past 20 years, making diabetes one of the fastest growing health challenges [13]. By 2030, it is estimated that 578 million people will have diabetes, rising to 700 million by 2045 [3]. Our understanding of wounds at the cell and molecular level needs to expand so that novel and low-cost therapies can be designed that might one day be able to heal most, if not all, wounds in a timely fashion and prevent amputation.

Macrophages are one of the key cells that regulate the wound repair process [14,15]. Wounds without macrophages have delayed re-epithelialisation, impaired angiogenesis, reduced collagen deposition, and reduced cell proliferation [16]. They are not a homogenous population of cells and several different combinations of phenotypes with distinct functions present at different times in the repair process [14,17,18]. Macrophage function is altered in people with diabetes such that they have a reduced capacity to clear an infection and their function in the later stages of repair is altered, leading to a delay in the repair process [14,19,20,21,22,23].

In this review, we will provide an overview of wound healing, the role of macrophages in inflammatory phase of wound healing, how these cells are altered by diabetes and by infection. We will also discuss some potential strategies to alter macrophages and inflammation in the wound to improve the repair process.

## 2. Overview of the Repair of an Acute Wound

Wound healing is complex and involves a series of coordinated and overlapping stages that need to come together for skin integrity to be restored [24,25]. These stages involve a range of distinct but often interlinked processes that include coagulation, inflammation, migration, proliferation, regeneration, and remodelling of the extra cellular matrix (ECM) [24,25]. These must all occur in a specific sequence, at a specific time and for a precise amount of time for an efficient and timely repair process. Delays in these steps can have detrimental outcomes, such as increased scar formation or the formation of non-healing wounds, and delays in closing the wound can lead to an increase in the risk of infection.

Wound healing begins first with haemostasis, followed by an inflammatory phase, a proliferative phase that leads to re-epithelization, and a remodelling phase, during which the scar matures [24,25]. While there is overlap in the phases, each phase is necessary for the next to be completed. In the first stage of the repair process, haemostasis stems the flow of blood from the damaged tissue. This takes place in seconds to minutes and is achieved by vasoconstriction and the formation of a clot [26]. To form the clot, platelets adhere to both the sub-endothelium that has been exposed by injury and to each other. They then degranulate and with the conversion of soluble fibrinogen to a network of insoluble fibrin, the platelet plug is stabilised [26]. This plug then acts as a provisional matrix for the inflammatory phase of wound healing to occur.

In the inflammatory phase, a number of immune cells, including tissue resident macrophages, and neutrophils and monocytes recruited to the site of injury from the blood, work with numerous cell types in and surrounding the injured skin to orchestrate the repair process [27,28]. This occurs through a combination of the actions of these cells themselves, their signalling pathways, and through the release of their soluble mediators, such as cytokines, chemokines, growth factors, and metabolites that signal other cells to perform specific tasks, to produce the final newly formed tissue [24]. While these actions occur over the course of seconds to months depending on their precise roles in the repair process, they must all come together at the correct times for an efficient and timely repair process, the timing of which might vary depending on the size and depth of the wound.

The macrophages’ role in the inflammatory phase is to clear pathogens and cell debris and to regulate, either directly or indirectly, the next stage in the repair process, the proliferative stage, through their ability to secrete cytokines and growth factors that stimulate keratinocytes, fibroblasts, and endothelial cells to proliferate, differentiate, and migrate [23,29]. This culminates in a new extracellular matrix (ECM), which allows cells to migrate over and re-epithelialise the wound and for new blood vessels to form and populate the wound. During the remodelling phase, macrophages secrete enzymes that remodel and alter the structure of the ECM and the wound [23,29].

## 3. The Inflammatory Phase of Wound Healing and Macrophages

Many different immune cells are recruited to the site of injury over the course of wound healing. In the initial stages after injury, prior to clot formation, there is some leakage of immune cells into the wound through areas of microhaemorrhaging and this small wave of immune cells then acts together with tissue resident macrophages as pioneer immune cells in the repair process [30]. Once blood flow is stemmed, immune cells must be attracted to and crawl over the plug fibres that act as a meshwork to populate the wound area. They are attracted by soluble factors released from the injured tissue and also after degranulation of the platelets within the clot [25]. Vascular permeability is induced by the release of histamine and other factors from activated mast cells [31]. The first recruited cell type seen in the wound is the neutrophil [28]. Their role is to phagocytose necrotic cells and damaged extracellular matrix, and to remove by phagocytosis any foreign material (e.g., pathogens) that may have entered the wound, releasing reactive oxygen species and secreting granule contents [28]. The recruitment of these neutrophils tends to peak around day 2 post-wound onset. From here, their numbers begin to decrease as these cells apoptose as part of the phagocytic process and are efferocytosed by macrophages to remove them from the wound, or they move away from the wound [24].

Studies in mice show that wound monocyte/macrophage numbers dramatically increase in an acute wound and then remain high from day 2 until around day 5, although this timing is wound-size dependent, with larger wounds taking longer [30]. The numbers then begin to decrease as re-epithelialisation occurs, falling to relatively low levels by day 7 and then returning to steady-state levels by day 14 [30]. Over this day 2–7 period, these cells have many different functions and so are responsible for a wide range of effects: they contribute to the initiation of inflammation but also resolve inflammation; they remove any pathogens that may have entered the wound; they clear up the neutrophils containing dead and partially digested microbes which have apoptosed and the remaining cell and ECM debris; they orchestrate the repair process through their secretion of cytokines and other factors, such as TGF- β1 and VEGF, that promote angiogenesis and attract fibroblasts that secrete extracellular matrix components necessary to rebuild the tissue; and they secrete enzymes that play a role in remodelling the ECM [15,24].

### Source and Plasticity of Wound Associated Macrophages

The monocytes/macrophages observed in wounds are derived from a number of sources. Initially, they consist of the tissue resident macrophages located in the skin prior to injury, of which there are two kinds; in the epidermal layer, these are predominantly Langerhans cells; in the dermal layer, they are mainly dermal macrophages [17]. The exact role of Langerhans and dermal macrophages in wound healing is currently unclear [32,33]. For people with diabetes, the number of these cells may be higher as there is already a significant increase in the numbers of these tissue resident macrophages in skin prior to injury compared to healthy people without diabetes [34,35].

Next on scene is the early wave of monocytes, which enter through microhaemorrhages caused by blood vessels damaged during the injury itself [30]. Factors released from platelets and in the wound environment trigger their differentiation into macrophages [36]. In the wound, these cells would also be exposed to alarmins, which include danger-associated molecular patterns (DAMPs) such as HMGB-1 or ATP released from cells after tissue damage, that activate these cells to become M1 macrophages [37]. Once activated, they can contribute to the initial proinflammatory phase and help recruit bone marrow-derived monocytes from the circulation 24 h later in a process that involves the chemokine receptor present on macrophages, CX3CR1 [30]. To accommodate the recruitment of a significant number of monocyte cells from the blood, there is an increase in myeloid lineage committed multipotent progenitors and monocytes in bone marrow that results in a 70% increase in monocytes in circulation on day 2, with their levels returning to steady-state levels after around day 4 [30,38]. These monocytes are classified as either classical/pro-inflammatory monocytes that are CD14^+^CD16^−^ capable of differentiating into pro-inflammatory M1 macrophages or anti-inflammatory monocytes that are CD14^low^/CD16^+^ that give rise to mostly M2 macrophages [27]. In mouse models of wound repair, circulating monocytes can also be divided into two groups: CX3CR1^low^CCR2^+^ Ly6C^+^ and CX3CR1^high^CCR2^−^Ly6C^−^ [15]. The first group produces pro-inflammatory cytokines and is the first to enter a wound, with the second entering later [15]. Factors in the local wound environment stimulate these bone marrow-derived monocytes to differentiate into macrophages and the precise combination of these factors is what appears to dictate macrophage phenotype, although the original phenotype of the bone marrow monocytes may also dictate the macrophage phenotype. In addition, some immune cells can proliferate in the wound [39]. It is the inflammatory monocytes/macrophages derived from the circulating monocytes, but not the mature wound macrophages, that are able to proliferate in the wound and, at the mid-stages of healing, these cells constitute around 25% of macrophage population.

This process is further complicated when analysing wounds because these macrophages can adopt different phenotypes in the wound [15,18,24,40]. Broadly speaking, phenotypes can be characterised by their function, with pro-inflammatory macrophages that are often described as M1 macrophages, and anti-inflammatory and reparative macrophages often termed M2 macrophages, although there are problems with categorising these cells this way and issues with the nomenclature [41,42]. In the early stages of repair, around 85% of macrophages in the wound have an M1 pro-inflammatory phenotype switching to around 80–85% anti-inflammatory M2 macrophages by days 5–7. The macrophage phenotype is complex and there are thought to be at least four types of pro-reparative M2 macrophages, namely M2a, M2b, M2c, and M2d, classified based on markers and functions, but not all appear to play a role in the wound [43]. How these different M2 phenotypes are derived is not yet fully elucidated. For example, are they derived from monocytes recruited at different points in the repair process with different phenotypes that then go on to differentiate into macrophages with specific phenotypes? Is it the changing environment that affects the phenotypes of the recruited cell? Alternatively, do the M1 macrophages in the wound differentiate into M2 due to environmental factors? Our understanding of the processes involved in wound formation is further complicated by the observation that these phenotypes appear not to be distinct. Instead, there is a continuum of phenotypes that evolves as the wound matures and is dependent on the cues in the environment.

Proinflammatory M1 macrophages have a high phagocytic capacity compared to un-activated macrophages (M0). They are typically found early on in the repair process and secrete cytokines such as interleukin 6 and TNF, the bactericidal molecule inducible nitric oxide synthase (iNos), as well as other mediators that regulate the early stages of healing. Macrophages with M2-like markers appear from day 1, although at this stage, the M1 phenotype seems to be dominant in the wound [15,18,24,40]. The switch to a M2 phenotype observed in in vitro models of wounds using human cell lines occurs after stimulation with IL-4/IL-13, but mouse wounds do not appear to contain these cytokines, and thus, another process must occur [18]. Plus, as discussed above, whether the more M2-like macrophages develop from newly recruited M0 or M1 macrophages already in the wound is not understood. It is known that wound macrophages can switch from pro-inflammatory to anti-inflammatory M2 phenotypes after efferocytosis, in which they remove apoptotic cells [18]. Wounds do have a high number of neutrophils and endothelial cells that have apoptosed as part of the repair process, and so their efferocytosis might allow for the large shift in phenotypes seen in the later stages of repair. Interestingly, M2 macrophages also retain their phenotypic plasticity such that if they are stimulated with LPS they can be pushed towards a more M1 phenotype, suggesting that an infection might alter their phenotype [44]. This is a complex process, and despite the significant progress made in the last two decades, it is unsurprising that there are still more questions to be answered on how these phenotypes form during the repair process and how they might be manipulated to improve the repair process.

## 4. Macrophage Dysregulation and the Repair Process

One common factor in all ulcers, both diabetic and non-diabetic, along with the inability to heal, is the dysregulation of the inflammatory phase of wound healing [23]. In diabetes, this is compounded by the fact that high levels of glucose seen in diabetes alters cells of the immune system, including macrophages, one of the key orchestrators of the repair process and defence against infection [21,45,46,47]. This dysregulation potentially contributes further to non-healing wounds and to the increased risk of infections. In individuals with diabetes, M1 macrophages drive the elevated and prolonged non-resolving inflammatory phase seen in DFUs [48]. Approximately 80% of cells at the chronic wound margin are pro-inflammatory M1 macrophages and there are compelling data from mouse and human studies to suggest that the shift to M2-like phenotypes may not proceed as expected, despite this shift being necessary for the repair process to progress [47,48].

Wound macrophages collected from individuals with diabetes patients and mice show M1 macrophages have high levels of pro-inflammatory molecules such as TNF, interleukin (IL)-1 β, and matrix metalloproteinase 9 (MMP9), and relatively low levels of the anti-inflammatory molecules and cytokines, such as TGF-β, IGF-1, and IL-10, typically associated with the repair phase [49,50]. The persistent inflammation and increased numbers of M1 macrophages in these wounds lead to reduced proliferation and migration of keratinocytes, fibroblasts, and endothelial cells necessary for the wound to repair. Both TNF and IL-1β have been shown to be key pro-inflammatory mediators that perpetuate the positive feedback loop that sustains inflammation in the diabetic wound [50,51]. Using db/db mice, a model of type 2 diabetes, it has been shown that topical application of an IL-1β neutralising antibody in wounds promotes the switch to more M2-like macrophages, increases growth factor numbers and accelerates re-epithelialisation [50]. Similarly, using another model of T2 diabetic mice, ob/ob mice, it has been found that the systemic application of TNF neutralising antibodies reduces wound inflammation by reducing macrophage numbers and increasing re-epithelialisation, leading to an improvement in the timing of the repair process [51]. Thus, a reduction in wound inflammation through the neutralisation of TNF and IL-1β can improve wound healing.

M1 macrophages in the wound also secrete large quantities of proteases including MMPs, such as MMP9 mentioned above, which cleave the ECM, often causing more damage to the wound and removing the newly laid ECM so that cells no longer have a scaffold to migrate over to repair the wound [52]. Cleaved ECM can also act as a chemoattractant as well as altering immune cell activity by shaping immune cell activation, differentiation, and survival [52]. Collectively, this leads to the attraction of more macrophages in the wound, which increases the inflammation that perpetuates the chronicity of wounds.

### Macrophage Function Is Altered in People with Diabetes

What is happening to the macrophage to cause this dysregulation of the repair process in DFUs? A number of factors contribute to the altered macrophage phenotypes, such as infection in the wound, high glucose, and advanced glycosylation end products (AGEs). Sustained exposure to high glucose levels in vitro produces macrophages that have reduced phagocytic activity, and therefore a reduced ability to clear an infection, reduced nitric oxide production, and cells that secrete more proinflammatory cytokines when stimulated [53,54]. Similar experiments with macrophages sourced from mice and individuals with diabetes show these macrophages are hyperresponsive to inflammatory stimulants and therefore secrete more pro-inflammatory cytokines, have difficulties switching to the more reparative M2-like phenotypes and, when looked at within the wound setting, prolong the inflammatory phase (Figure 1) [19,45,47,50,55].

This switch in macrophage phenotype is essential for a timely repair process. Increasing the levels of M2 macrophages in wounds leads to an increase in cells secreting the anti-inflammatory cytokines that dampen inflammation and an increase in the growth factors necessary for proliferation, migration, and the repair process [24]. The importance of this switch has been seen in many studies indicating that, depending on the phenotype and the timing of their actions, macrophages may have a detrimental or positive effect, either damaging tissue or aiding the repair process. Studies using diphtheria toxin-mediated macrophage depletion models in mice show that M2 macrophages (CD206^+^/CD301b^+^) are critical for activation of reparative processes during the mid-stages of wound healing [56]. Blocking the switch to M2 macrophages in wounds using the inhibitor GW2580, which blocks the CSF-1 signalling cascade that drives macrophage differentiation, proliferation, and survival, results in persistent inflammation, less collagen, and increased M1 macrophages in the wound [57], while transplanting day 5 wound M2 (CD206^+^/CD301b^+^) macrophages to day 3 wounds in mice leads to significantly increased proliferation and fibroblast repopulation [57]. Collectively, these and other studies show that this M1 to M2 switch is important to a timely repair process and that the impaired switching to M2 phenotypes seen in diabetic wounds is associated with poor angiogenesis, decreased collagen deposition, and reduced wound closure (Figure 1) [55].

Hyperglycaemia leads to an increase in advanced glycation end products (AGEs), which are proteins or lipids glycated due to their exposure to sugars [54]. The diabetic wound environment accumulates both AGEs and macrophages that express high numbers of the receptor for AGEs (RAGE). In vitro, the addition of AGEs to M1 macrophages reduces their ability to phagocytose. The phagocytosis of apoptosed cells (efferocytosis) is a key process that activates the switch to M2 macrophages. The disruption of this process greatly impacts repair by keeping wounds open longer and increasing the risk of infection. When anti-RAGE antibodies are applied to wounds in vivo, there is an increase in phagocytosis (efferocytosis) of neutrophils and a push towards a switch of macrophages to M2 phenotype, leading to improved healing, suggesting that high levels of AGEs in the wound impact the switch in phenotype and timely repair [58]. In vivo macrophages taken from the wounds of diabetic mice exhibit impaired efferocytosis, leading to an increase in the wound of apoptotic cells, both neutrophils and endothelial cells [47,59]. This increase in apoptotic cell load is seen in the wound tissue of individuals with diabetes when compared to the levels in those without [47]. To test whether this increase in apoptotic cells might alter wound healing, anti-CD95 (clone JO2), which induces apoptosis, was used to increase the apoptotic burden in diabetic mice [47]. The results showed that the increase in apoptotic cells delayed wound closure [47]. The increase in apoptotic cells also resulted in an increase in the pro-inflammatory cytokines TNF and IL-6, concomitant with a reduction in the anti-inflammatory cytokine IL-10 [47]. This suggests that an increase in apoptotic cells can influence the wound repair process.

In vitro models show that two cytokines, IL-4 and IL-13, are able to induce the switch of macrophages to the M2 phenotype. However, using sponges placed in the wound of non-diabetic mice to collect wound fluid, neither cytokine was present, suggesting that they may not be the driver of M2 phenotypes in wounds [18]. Efferocytosis of the apoptosed cells pushes macrophages towards an M2 phenotype and this might be the main driver of the switch in wounds [24]. The reduced ability of macrophages in people with diabetes to efferocytose appropriately could be the cause of the reduced M2 phenotype and contribute in part to the increased presence of M1 macrophages.

People with diabetes are highly susceptible to infections, further complicating the slow healing rate of DFUs [60]. In certain instances, patients with DFUs can require hospitalisation and in some cases amputation as a final resort in their treatment. DFU infection leads to a 50-fold increased risk of hospitalisation, with around 5% of patients with an infected DFU needing a major amputation and 20–30% a minor amputation [61]. A number of factors can play a response to infection, including the microbial load, biofilms, and the range of bacteria present [59,60]. Bacteria such as the gram-positive *Staphylococcus aureus*, *Streptococcus agalactiae*, *Streptococcus dysgalactiae, Enterococcus* spp., gram-negatives such as *Pseudomonas aeruginosa, Escherichia coli*, *Klebsiella* spp., *Proteus* spp., and anaerobes such as *Bacteroides* spp. and *Peptostreptococcus* spp. have been found in DFUs [62,63]. DFUs are thought to have higher microbial loads than venous leg ulcers [63]. The diversity of these loads also impacts wound healing. DFUs with mild or moderate diabetic foot infections (DFIs) contain predominantly *Staphylococcus aureus* and Streptococcus genera, while severe DFIs have increased diversity comprising both aerobes and anaerobes including *Proteus* spp., *Prevotella bivia, Porphyromonas* spp., *Enterobacterales,* and *Anaerococcus* spp. such as *Anaerococcus lactolyticus* [62]. This increase in diversity leads to an upregulation in host genes in the wound tissue related to inflammation when compared to mild to moderate DFI gene levels, while genes related to epithelial—mesenchymal transition, which are required for the wound to close, are reduced [62]. Thus, it appears that the number and diversity of infecting organisms can alter the host immune response [27]. Whether this change in host response is coming from the already altered macrophage response in these wounds is not clear. The reduced phagocytosis by these cells would reduce their ability to clear an infection, necessitating antibiotics or other treatment measures.

*Pseudomonas aeruginosa* commonly infects diabetic wounds and it has been shown in mice models that infection with this bacterial species leads to an increased recruitment in macrophages [64]. A greater number of macrophages in an infected environment means that more of these cells are classically activated, which means that they will switch to the M1 phenotype, thus increasing the numbers of M1 macrophages in the wound. These cells in turn recruit more macrophages and the cycle continues. Consequently, in the infected wound, there is an excessive activation of M1 macrophages, the switching of macrophages phenotype is altered so the wound cannot close and there is a further delay in wound healing [64]. The reduced ability of cells to phagocytose pathogens effectively impacts not only the repair process but also the ability to clear an infection.

## 5. Concluding Remarks

The dysregulation of macrophage functions in diabetic individuals, such as the reduced ability to phagocytose pathogens and apoptosed cells, leads to a reduced ability to switch to the more reparative M2 phenotype. This results in an increase in the number macrophages being recruited to the wound, and their activation to become M1 macrophages. With few M2 macrophages present this leads to a delay in wound closure and in many cases further damage to the tissue. Further compounding this is the reduced ability to clear an infection, increasing the risk of amputation as the ultimate treatment to DFUs.

There are many questions left to answer before we fully understand the repair process, such as what are the exact phenotypes needed in DFUs to promote effective wound healing and can we alter macrophages to adopt those phenotypes? As our understanding of what is happening to macrophages in wounds increases, so will the opportunities to design and test new potential therapies that might complement existing treatments in future.

It is unclear exactly what control of wound inflammation might look like. It could be by the use of biomaterial dressings that might modulate the immune system or through the modulation of the skin’s own host defence peptides to clear an infection and alter the inflammatory phase [65,66,67,68,69]. It might be as simple as reducing the number of monocytes/macrophages entering the wound by repurposing of drugs, for example the use of anti-integrin antibodies to reduce the macrophage load [70]. Alternatively, it might be altering the phenotype of the macrophage, reducing the factors such as TNF in the wound with an anti-TNF antibody to reduce inflammation, or the design of new therapies to dampen wound inflammation [51]. This would be in addition to wound dressings, potential debridement, offloading, and control of factors such as hyperglycaemia, and preventing and treating any infection that might occur in the wound.

## Figures and Tables

**Figure 1 molecules-26-04917-f001:**
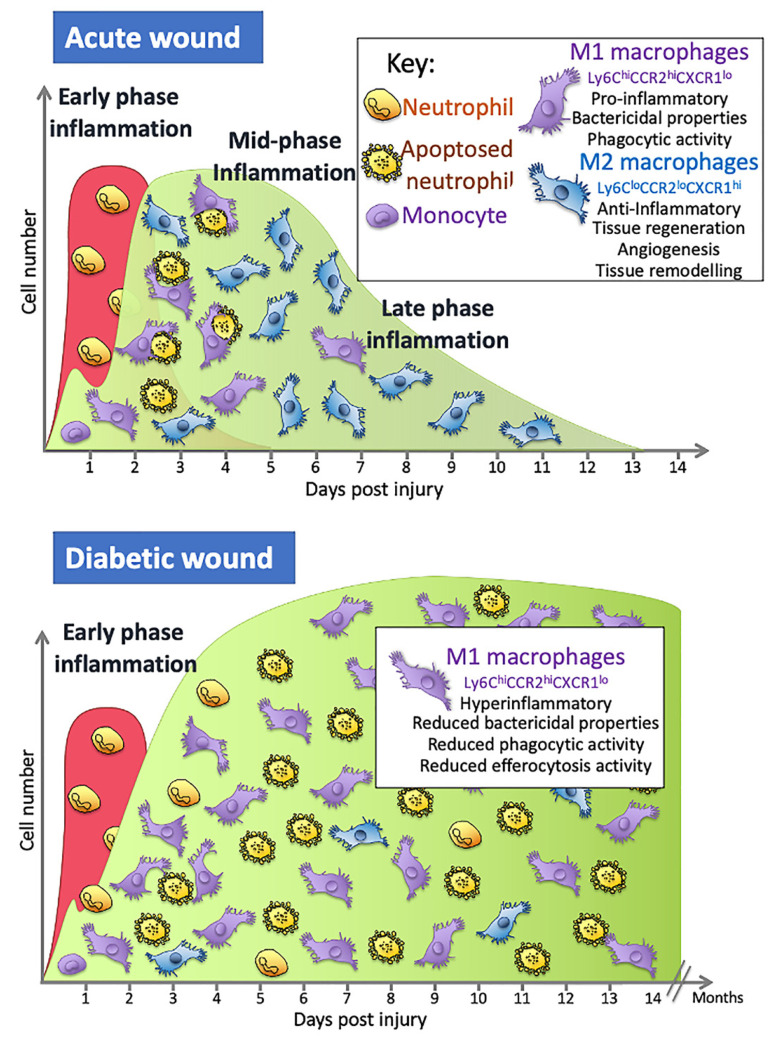
During the early stages of inflammation in an acute wound, the wound is predominantly populated with M1 macrophages that are pro-inflammatory. A switch in the predominant phenotype is seen later in the inflammatory phase, where the majority of macrophages are M2. In wounds of people with diabetes, macrophage numbers are altered and secrete more pro-inflammatory cytokines, have reduced phagocytic and efferocytosis abilities and are less likely to switch to the M2 phenotype in the mid- to late-inflammation phase that is as seen in normal acute wounds.

## Data Availability

Not applicable.

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
