# Peer review of "Skin Wound Healing: Normal Macrophage Function and Macrophage Dysfunction in Diabetic Wounds"

_molecules, 2021, doi:10.3390/molecules26164917_

Round 1
Reviewer 1 Report
Editor and Reviewer comments:
The manuscript entitled "Macrophage (dys)function in normal and diabetic skin wounds" by Savannah M Aitcheson et al., reviewed the macrophage dysregulation in DFUs and discussed these cells about the inflammatory phase of wound healing and repair process. Non-healing diabetic foot ulcer (DFU) is a clinical and economic needs for better wound therapies. However, many of these wounds still fail to heal and the amputation is required. The molecular-level needs to be further understanding the aspects of wound healing. The opportunities to design and test new potential therapies that might complement existing treatments in future, if we know what is happening to macrophages in wounds.
Some points are inadequate and some details that are flawed. To improve this manuscript, several issues should be addressed as follows:
- The authors should add more new references with your work, especially in the introduction (Part 1). Few literatures were cited in the past three years. Please explain in detail why macrophage can play a prominent role in the wound repair process, in addition, more discussion about macrophage in the wound healing should be added to the introduction.
- Please explain the core reason in detail why macrophage is selected for the repair process of wound healing, the point in the review is few.
- The authors should give the detailed steps to the inflammatory phase.
- The discussion section is too long for the reader. The author should integrate the references with the discussion section and delete the irrelevant sentences that are repeatedly described in the discussion section.
- I suggest that the author add more transitional sentences between sections and between paragraphs in each section, and pay more attention to the theme of this review in the introduction, while in the conclusion, besides echoing the theme of the introduction, the author should also take consideration to the relevance of the whole manuscript.
- The authors should add the limitations of the review and the future work in the conclusions section.
Author Response
Thank you for your review, it was helpful in improving the manuscript.
Point 1: The authors should add more new references with your work, especially in the introduction (Part 1). Few literatures were cited in the past three years. Please explain in detail why macrophage can play a prominent role in the wound repair process, in addition, more discussion about macrophage in the wound healing should be added to the introduction.
Response 1: Some newer references have been added and a section explaining why we are looking at macrophages and their prominent role in wound healing has been added to part 1. (Lines 70-81).
Point 2: Please explain the core reason in detail why macrophage is selected for the repair process of wound healing, the point in the review is few.
Response 2: This has now been added to the (Lines 70-81)
Point 3: The authors should give the detailed steps to the inflammatory phase.
Response 3: The steps are given in part 2 (lines 107-123) and this is expanded, which particular reference to macrophages, in section 3 Lines 126-260.
Point 4. The discussion section is too long for the reader. The author should integrate the references with the discussion section and delete the irrelevant sentences that are repeatedly described in the discussion section.
Response 4: the repeated sentences have been deleted or altered to remove repetition (lines 303-306, 347, 410-411).
Point 5: I suggest that the author add more transitional sentences between sections and between paragraphs in each section, and pay more attention to the theme of this review in the introduction, while in the conclusion, besides echoing the theme of the introduction, the author should also take consideration to the relevance of the whole manuscript.
Response 5: This has been fixed up through the manuscript (tracked changes on)
Point 6: The authors should add the limitations of the review and the future work in the conclusions section.
Response 6: The questions to be answered and the potential future work has been added to the conclusion section.
Reviewer 2 Report
The manuscript “Macrophages (dys)function in normal and diabetic skin wound” by Savannah Aitcheson with co-authors provides a review of the normal macrophages function in wound healing and macrophages dysfunction in diabetic wound healing.
Overall, the manuscript is well organized, well written and of interest for the readers.
Specific comments
- The title is difficult to understand, and it should be corrected. What is the normal skin wound? Do macrophages dysfunction in normal wound? I suggest to change the title like: Skin wound healing: normal macrophages function and macrophage dysfunction in diabetic wounds.
- There are several repeats in the text. Lines 253-259 (page 7) is repeat of lines 116 (page3) -128 (page 4). Lines 281-283 (page 8) is repeat of the lines 193-200 (page 5). Line 327 (page 9)- line 330 (page 10) is repeat of the lines 244-247 (page 7). Combine all repeated information early in the text and refer to it later.
- Line 319. Provide examples of aerobic and anaerobic bacteria that infect diabetic wounds.
Minor comments
- Line 17 ”The macrophages that coordinate this repair process are complex”. Do you mean the macrophage population is diverse? Or macrophage function is complex? Please clarify it.
- Line 39 “This and other factors negatively affect …” What are other factors?
- Line 120 “During this time these cells…” Do you mean that during the whole period of the wound healing macrophages play wide variety of the roles?
- Line 123 “the spent neutrophils...” Do you mean apoptotic neutrophils?
- Line 160 “the phenotype at the start may also play a factor ...” Do you mean phenotype also plays a role?
- Line 160 “some immune cells can proliferate in the wound.” Do you mean some macrophages or other types of the cells like T-cells can proliferate? Clarify.
- Line 173 The question “For example, are they derived from monocytes…” is too long and must be separated into 2-3 different questions.
- Line 237 “perpetuate the chronicity of wound.” Do you mean Infiltration of additional macrophages maintain chronic wound?
- Line 306 “The inability of macrophage in diabetic people ...” change to “reduced ability”
- Line 339 “Given that diabetes itself impairs wound healing, the inability of cells…” How does diabetes itself impairs wound healing? Change “inability” to “reduced ability”.
- Line 344 “The dysregulation of macrophage function, combined with inability to switch phenotype…” What macrophage function is dysregulated? Clarify. Change “inability” to “reduced ability”.
Author Response
Thank you for your review, it was helpful in improving the manuscript.
Response to Specific comments.
Point 1: The title is difficult to understand, and it should be corrected. What is the normal skin wound? Do macrophages dysfunction in normal wound? I suggest to change the title like: Skin wound healing: normal macrophage function and macrophage dysfunction in diabetic wounds.
Response 1: Thank you for the suggestion. This has now been altered (lines 2-3).
Point 2: There are several repeats in the text. Lines 253-259 (page 7) is repeat of lines 116 (page3) -128 (page 4).
Response 2: The text has now been altered to ‘This switch in macrophage phenotype is essential for a timely repair process. Increasing the levels of M2 macrophages in wounds leads to an increase in cells secreting the anti-inflammatory cytokines that dampen inflammation and an increase in the growth factors necessary for proliferation, migration and the repair process.” (lines 312-315).
Point 3: Lines 281-283 (page 8) is repeat of the lines 193-200 (page 5).
Response 3: The text has now been deleted (Lines 355).
Point 4: Line 327 (page 9)- line 330 (page 10) is repeat of the lines 244-247 (page 7). Combine all repeated information early in the text and refer to it later.
Response 4: The text has now been altered to say ‘The reduced phagocytosis by these cells would reduce their ability to clear an infection necessitating antibiotics or other treatment measures[63].’ (Lines 416-417).
Point 5: Line 319. Provide examples of aerobic and anaerobic bacteria that infect diabetic wounds.
Response 5: These have now been added to the text (lines 390-410).
Point 6: Line 17 ”The macrophages that coordinate this repair process are complex”. Do you mean the macrophage population is diverse? Or macrophage function is complex? Please clarify it.
Response 6: The text has now been altered to say ‘The macrophages that coordinate this repair process are complex: they originate from different sources and have distinct phenotypes with diverse functions that act at various times in the repair process.’ (Lines 18-20).
Point 7: Line 39 “This and other factors negatively affect …” What are other factors?
Response 7: The text has now been altered to say ‘This and other factors such as social isolation and depression negatively affect patients’ perceived health-related quality of life.’ (Lines 47-49).
Point 8: Line 120 “During this time these cells…” Do you mean that during the whole period of the wound healing macrophages play wide variety of the roles?
Response 8: This has been clarified in the text’ ‘Over this day 2-7 period’ (line 150)
Point 9: Line 123 “the spent neutrophils...” Do you mean apoptotic neutrophils?
Response 9: Spent neutrophils refers to neutrophils that have performed their role and contain dead or partially digested microbes. After this, they apoptose. ‘Spent’ has now been removed and the text altered to state ‘they clear up the neutrophils containing dead and partially digested microbes which have apoptosed”. (Line 153-154).
Point 10: Line 160 “the phenotype at the start may also play a factor ...” Do you mean phenotype also plays a role?
Response 10: The text has now been altered to say ‘although the original phenotype of the bone marrow monocytes may also dictate the macrophage phenotype.’ (Lines 195-196).
Point 11: Line 160 “some immune cells can proliferate in the wound.” Do you mean some macrophages or other types of the cells like T-cells can proliferate? Clarify.
Response 11: It is the monocytes and macrophages. This has now been clarified in the text ‘It is the inflammatory monocytes/macrophages derived from the circulating monocytes, but not the mature wound macrophages, that are able to proliferate in the wound and, at the mid-stages of healing, these cells constitute around 25% of macrophage population.’ (Lines 196-199).
Point 12: Line 173 The question “For example, are they derived from monocytes…” is too long and must be separated into 2-3 different questions.
Response 12: This has been fixed: ‘For example, are they derived from monocytes recruited at different points in the repair process with different phenotypes that then go on to differentiate into macrophages with specific phenotypes? Is it the changing environment that affects the phenotypes of the recruited cell? Alternatively, do the M1 macrophages in the wound differentiate into M2 due to environmental factors [38]? (Lines 211-215).
Point 13: Line 237 “perpetuate the chronicity of wound.” Do you mean Infiltration of additional macrophages maintain chronic wound?
Response 13: The text has now been altered to ‘which increases the inflammation that perpetuates the chronicity of wounds.’ (Lines 297).
Point 14: Line 306 “The inability of macrophage in diabetic people ...” change to “reduced ability”. Line 339 “Given that diabetes itself impairs wound healing, the inability of cells…” How does diabetes itself impairs wound healing? Change “inability” to “reduced ability”. Line 344 “The dysregulation of macrophage function, combined with inability to switch phenotype…” What macrophage function is dysregulated? Clarify. Change “inability” to “reduced ability”.
Response 14: All have been changed to ‘reduced ability’ as suggested (Lines 381, 416, 426).
Round 2
Reviewer 2 Report
Authors provided responses to all my comments, and significantly improve a manuscript. I recommend to accept the manuscript for publication.